# Discovery of Novel Diphenyl Acrylonitrile Derivatives That Promote Adult Rats’ Hippocampal Neurogenesis

**DOI:** 10.3390/ijms25021241

**Published:** 2024-01-19

**Authors:** Si-Si Liu, Cong-Xuan Ma, Zheng-Yang Quan, Jing Ding, Liang Yang, Si-Meng Liu, He-Ao Zhang, Hong Qing, Jian-Hua Liang

**Affiliations:** 1Key Laboratory of Medicinal Molecule Science and Pharmaceutical Engineering, School of Chemistry and Chemical Engineering, Beijing Institute of Technology, Beijing 102488, China; lsshbnu@163.com (S.-S.L.); 3120195674@bit.edu.cn (C.-X.M.); 3120205655@bit.edu.cn (J.D.); dmz2015cpu@163.com (S.-M.L.); 2School of Life Science, Beijing Institute of Technology, Beijing 100081, China; 3120215692@bit.edu.cn (Z.-Y.Q.); lyang@yau.edu.cn (L.Y.); zhangheao@bit.edu.cn (H.-A.Z.)

**Keywords:** neurogenesis, proliferation, differentiation, SARs, diphenyl acrylonitrile

## Abstract

We previously discovered **WS-6** as a new antidepressant in correlation to its function of stimulating neurogenesis. Herein, several different scaffolds (stilbene, 1,3-diphenyl 1-propene, 1,3-diphenyl 2-propene, 1,2-diphenyl acrylo-1-nitrile, 1,2-diphenyl acrylo-2-nitrile, 1,3-diphenyl trimethylamine), further varied through substitutions of twelve amide substituents plus the addition of a methylene unit and an inverted amide, were examined to elucidate the SARs for promoting adult rat neurogenesis. Most of the compounds could stimulate proliferation of progenitors, but just a few chemicals possessing a specific structural profile, exemplified by diphenyl acrylonitrile **29b, 32a,** and **32b**, showed better activity than the clinical drug **NSI-189** in promoting newborn cells differentiation into mature neurons. The most potent diphenyl acrylonitrile **32b** had an excellent brain AUC to plasma AUC ratio (B/P = 1.6), suggesting its potential for further development as a new lead.

## 1. Introduction

It has been established that hippocampal neurogenesis in rodents, non-human primates, and humans persists throughout the life span [1], but it drops sharply in aged adults and Alzheimer’s Disease patients [2,3]. A positive correlation exists between higher numbers of neuroblasts and an enhanced cognitive status [2]. Therefore, regulating adult mammalian hippocampal neurogenesis through small molecules is a potential strategy for the treatment of neurodegenerative disorders [4]. Many synthetic potential scaffolds have been identified, such as 4′-dimethylamino-7, 8-dihydroxyflavone [5], N-acetyl bioisosteres of melatonin [6], ISX-9 [7], 2-phenylamino-quinazoline [8], pyrrolidinones [9], melatonin–pinoline hybrids [10], isoquinoline-1,4(6H)-dione [11], neurodazine (Nz), and neurodazole (Nzl) [12], which were documented in our recent review [4]. However, small molecules that are efficient in vivo are still urgently needed to evaluate their potential in the treatment of neurodegenerative diseases.

Considering the presence of the blood–brain barrier (BBB) and the cell type complexity required for neurogenesis in the hippocampus, devising simple, predictive assays has proven difficult, and little consensus exists regarding which receptors or enzymes represent ideal targets [13]. For this reason, conducting in vivo phenotype screening is essential to examine their potential in promoting neurogenesis. As a result, **P7C3** and **NSI-189** (Figure 1) were demonstrated to be two of the most promising molecules. **P7C3** showed a great potential in treating a variety of neurodegenerative diseases, while **NSI-189** is in Phase II as a monotherapy in major depressive disorder [14,15,16,17,18,19]. It is noted that **P7C3** and **NSI-189** have different modes of action. **P7C3** is a neuroprotective agent (by lowering the death rate of newborn cells), although the researchers initially intended to search for molecules with a capacity to stimulate proliferation [13,20]. In other words, a significant increase in newborn cells in the hippocampus exposed to **NSI-189** was observed after a short time, while the significance when exposed to **P7C3** was observed after a long time [15].

To produce new mature neurons in the hippocampal dentate gyrus (DG) region, both the proliferation of progenitors and differentiation into neurons are key steps for neurogenesis. Unfortunately, the structure–activity relationships (SARs) of the stimulators are less investigated at the stage of differentiation [4]. It is unclear which pharmacophore could determine the fate of the newborn progenitors in an adult mammalian brain. We previously discovered *N*-*trans*-3′,4′-methylenedioxystilben-4-yl acetamide (coded **WS-6**, Figure 1) based on natural pterostilbene [21]. **WS-6** showed an excellent capacity in promoting the proliferation of progenitors and differentiation into neurons in rodents’ hippocampi, which possessed the same mode of action as that of **NSI-189** [21]. Later, **WS**-**6** proved to be a new antidepressant in correlation to its function of stimulating neurogenesis [22]. 

We herein report SARs of stilbene analogs both at the proliferation stage and the differentiation stage. Thus, we noted that only a few of the chemicals with the specific structural profile have a further capacity to promote newborn cell differentiation into mature neurons. 

## 2. Results and Discussion

### 2.1. Exploration of SARs Based on WS-6

**WS-6** [21] was used as the lead compound for further structural modification and optimization. We initially examined the impact of substituents such as various amides, a fluorine atom, a methylenedioxy group, and a 3,5-dimethoxy group on the activity (Figure 1). To evaluate the influence of linker length on the activity of the scaffold, we designed two diphenyl propene skeletons (Figure 2). We incorporated a cyano group or a tertiary nitrogen linker (Figure 3 and Figure 4) to adjust the lipid–water partition coefficient, which was a crucial factor in determining the ability of small molecules to penetrate the blood–brain barrier. To further develop new substituents, we designed twelve different amides or sulfonamides (illustrated in Figure 2). Finally, we introduced an inverted amide to the stilbene and inserted an additional methylene unit between the amide and the stilbene (Figure 5). 

### 2.2. Chemistry

The procedures to synthesize the stilbene **8**, **9**, and **12** are outlined in Figure 1. Commercially available 3,4-methylenedioxybenzaldehyde (**1**) or 3,5-dimethoxybenzaldehyde (**2**) was used to produce the corresponding styrene (**3** or **4**) by using a Wittig reaction. The Heck reaction of **3** or **4** with 1-iodo-4-nitrobenzene (**5**) or 2-fluoro-4-iodoaniline (**10**) led to the stilbene **6** or **11**. A reduction of **6** in the presence of SnCl_2_ was performed to yield **7**. The key intermediates **7** and **11** reacted with acylated reagents such as carboxylic acids, anhydrides, and acyl chlorides to give the target compounds **8** (**8c–8e**, **8g–8h**, **8j–8l**), **9b**, and **12f.**

The procedures to synthesize the diphenyl propene **18**, **23**, and **24** are outlined in Figure 2. The preparation of chalcone (**15**) began with the aldol condensation reaction of commercially available phenyl ethanone (**13**) with benzaldehyde (**14**), followed by reduction with NaBH_4_ and then dehydration to obtain diphenyl propylene (**17**). Regional isomerization of olefin (**22**) can be performed in a similar way using benzaldehyde (**1** or **2**) and phenylethyl ketone (**19**). The diphenyl propene **18b**, **23b**, **23i**, and **24b** were prepared through the acylation reaction of **17** or **22** with methanesulfonyl chloride or isobutyric anhydride.

The procedures to synthesize the diphenyl acrylonitrile **29** and **32** are outlined in Figure 3. A condensation reaction of commercially available benzaldehyde (**1**) and phenylacetonitrile (**25**) followed by the reduction of nitro to amino groups yielded **28**. Similarly, **31** was achieved from **14** and **26**. The diphenyl acrylonitrile **29a–29b** and **32a–32b** were obtained through acylation.

The procedure to synthesize the diphenyl trimethylamine **36** is outlined in Figure 4. Condensation and reduction of commercially available *N*-(4-formylphenyl)acetamide (**33**) and 3,4-methylenedioxymethanamine (**34**) yielded the intermediate **35**. Compound **36a** was obtained through methylation on the nitrogen atom.

The procedures to synthesize the target compounds **39b** and **42b** are outlined in Figure 5. The Heck reaction of **3** with commercially available **37** led to the acid **38**, and acylation of **38** with an amine produced the inverted amide **39b**. The amide installed onto the additional methylene group **42b** was obtained through a Heck reaction of **40** followed by acylation of **41**. 

### 2.3. Structure–Activity Relationships

Previous in vitro or in vivo studies mainly focused on stimulating the proliferation effect of the small molecules [21]. In this study, a two-step protocol was employed. First, the stimulating function of the compounds on progenitor cells in the rodents’ hippocampi was examined with using bromodeoxyuridine (BrdU) labeling. Next, the potent compounds were further screened to ascertain their ability to promote the differentiation of neural progenitor cells, as confirmed by dual labeling of BrdU and NeuN (neuronal nuclear antigen).

#### 2.3.1. Structure–Activity Relationships for Stimulating Proliferation

Adult rats were administered intraperitoneally with the prepared compounds at a dose of 4.0 mg/kg/day for a period of 28 days, followed by a BrdU pulse. After 24 h, the rats were sacrificed, and their brain tissues were isolated and cryosectioned. Immunohistochemical staining was performed and eight brain tissue slices were randomly selected for statistical analyzing [21].

To investigate the impact of amide pharmacophores on proliferative activity, a series of stilbenes, namely, **8c** (*N*,*N*-dimethylaminoformamide), **8d** (cyclobutylformamide), **8g** (benzylformamide), **8h** (pyridi-2-ylformamide), **8j** (*N*,*N*-dimethylsulfonamide), **8k** (cyclopropylsulfonamide), **8l** (benzenesulfonamide), **9b** (isobutyramide), and **12f** (benzamide) were designed. The data in Figure 3 and Figure 4 shows that both sulfonamides and amides can enhance the proliferation of brain progenitor cells. However, irrespective of an amide or a sulfonamide, the introduction of (hetero) aryl substituents (**8g**, **8h**, **8l**, Figure 3; **12f**, Figure 4) adversely affected the proliferative activity. Conversely, cyclic alkyl groups and aza-alkyl groups (**8d**, **8j**, **8k**, Figure 3) demonstrated a greater propensity for enhancing activity. 

In contrast to stilbenes, diphenylpropene **18b** and **23b** maintained the same proliferative activity as **9b** (Figure 4). To gain a deep understanding of the impact of diphenyl propene on proliferative activity, we substituted 3,4-methylenedioxy with 3,5-dimethoxy (**23b** vs. **24b**) or replaced amide with sulfonamide (**23b** vs. **23i**). The activity of **24b** remained the same, while the activity of methanesulfonamide **23i** decreased, which underscored the different structure–activity relationships in the diphenyl propene and the stilbene frameworks (Figure 5).

An attempt to bridge two phenyl rings using a trimethylamine linker (**36a**) instead of alkene linkers abolished the activity (Figure 5). The diphenyl acrylonitrile **29b** exhibited satisfactory activity. However, compound **8e** with a cyano group shifted from the linker to the amide substituent did not display significant activity. These findings corroborated that stilbene was the preferred scaffold with proliferative activity and was amenable to certain structural modifications. 

The amide inversion (**39b**) displayed activity comparable to that found in amides **8c**, **8d**, and **29b** (Figure 6) [21]. Unfortunately, the introduction of an additional methylene group between the amide and the stilbene led to a loss of activity (**42b**). These experimental findings emphasized the diverse SARs exhibited by amide substituents.

#### 2.3.2. Structure–Activity Relationships for Promoting Differentiation

In an effort to investigate the effects of various compounds on the differentiation fate of neural progenitor cells following proliferation, we selected a set of nine active compounds (**8c**, **8d**, **8j**, **8k**, **18b**, **23b**, **24b**, **29b**, and **39b**) and an inactive compound, **42b**. The study employed a rat model, wherein nine-week-old rats were administered intraperitoneally with the compound at a dose of 4.0 mg/kg/day for 28 days and then treated with a pulse of BrdU. Instead of sacrificing the rats 24 h after the BrdU injections, the rats were sacrificed after another 28 days. The brains were then isolated and cryosectioned. Eight randomly selected slices were subjected to immunohistochemical staining. This staining enabled the co-localization of the BrdU-incorporated cells with the NeuN marker, enabling the definition of the compounds that could promote the differentiation of newborn neural progenitor cells into mature functional neurons.

However, after a 28 day waiting period, only five compounds (**8c**, **8d**, **8k**, **29b** and **39b**) could promote the proliferation of progenitors (Figure 7A). Compounds **18b**, **23b**, and **24b**, featuring a diphenyl propylene scaffold, did not exhibit significant proliferative activity over the longer 28 day period. Regarding the amide and sulfonamide substructures, cycloalkyl substituents were more favorable than the aza-alkyl groups to enhance the proliferative activity (**8k** vs. **8j**, **8d** vs. **8c**). Compound **39b**, characterized by amide inversion, exhibiting proliferative activity within both the short and long durations, whereas compound **42b**, with the insertion of a methylene group between the amide and the benzene ring, remained devoid of proliferative activity. Unexpectedly, the analysis of BrdU and NeuN double staining revealed that only compound **29b** significantly increased the number of newborn neurons. Compound **39b** ranked second in terms of activity, without statistical significance (Figure 7B).

#### 2.3.3. Structure–Activity Relationships of **29b** Analogs

Inspired by the success of **29b**, we synthesized **29a**, an acetamide analog of compound **29b**. The corresponding isomers of **29a** and **29b**, **32a** and **32b**, varying the position of the cyano group, were also prepared. Because **NSI-189** was a stimulator for neurogenesis [17], we utilized **NSI-189** as a control. The results depicted in Figure 8 showed that both **29a** and **NSI-189** were inactive at a low dose of 4 mg/kg, while the other three compounds, **29b**, **32a**, and **32b**, exhibited proliferative activity after a day.

Next, we further examined the effects of **29a**, **29b**, **32a**, **32b**, and **NSI-189** on the differentiation fate of newborn progenitors (Figure 9). The trend in the differentiation stage was observed to be consistent with that in the proliferation stage. Among the compounds, isobutyramide was more favorable than acetamide in enhancing activity (**29b** vs. **29a**, **32b** vs. **32a**). The position of the cyano group played a crucial role in defining activity, because **32a** and **32b**, with a cyano substitution distal to the amide-substituted phenyl ring, are more active than **29a** and **29b**, with a cyano substitution adjacent to the amide-substituted phenyl ring. Compound **32b** showed the highest activity, while **29a** and **NSI-189** had no significant neurogenesis-promoting activity at the dose of 4.0 mg/kg. 

Finally, we performed cytotoxicity tests on compounds **29b**, **32a**, and **32b**. Among them, the IC_50_ values of compounds **29b** and **32b** were greater than 200 μM, and only compound **32a** had a weak inhibitory effect (IC_50_ = 109.29 ± 12.91 μM) on hCMEC/D3 cell lines (Appendix A).

### 2.4. Pharmacokinetic Studies

We conducted pharmacokinetic experiments (*n* = 3 rats per group) on the most promising compound, **32b**, to examine its pharmacokinetic parameters and distribution (Table 1). The maximum drug concentrations observed in the plasma and brain tissue of rats were 132 ng/mL and 168 ng/g, respectively. The exposure of the drug in brain tissue in terms of AUC (area under the curve) (i.e. the brain to plasma ratio, B/P) was 1.59 times higher than that in plasma (383 h*ng/mL in plasma and 608 h*ng/g in brain tissue). The half-life of the drug in brain tissue was found to be better than in plasma. Taken together, compound **32b** could effectively traverse the BBB and exhibited great overall uptake in the brain.

Calculating the logP (lipid–water partition coefficients) indicated 3.82 for acetamide **WS-6** and 4.66 for an isobutyramide analog of **WS-6**. However, ClogP of the isobutyamides **29b** and **32b** was comparable to **WS-6** (Clog P: 3.28 for **29b** and **32b**; 2.44 for acetamide **29a** and **32a**). Thus, the introduction of the cyano group played a role in regulating the lipid–water partition coefficient, which may account for the improved ability of **32b** to penetrate the BBB. 

## 3. Materials and Methods

### 3.1. Chemistry

All reagents and solvents were purchased from commercial sources (J&K Scientific) and used without further purification unless otherwise noted. All non-aqueous reactions were carried out under argon using dry solvents, unless otherwise noted. Reactions were monitored through thin-layer chromatography, and 254 nm UV light was used for visualization. Column chromatography was performed on silica gel (100–200 or 200–300 mesh) purchased from Qingdao Haiyang. ^1^H and ^13^C nuclear magnetic resonance (NMR) spectra were recorded on Bruker ARX 400/500 MHz with tetramethylsilane (TMS) as an internal standard. High resolution mass spectra (HRMS) were obtained with Agilent 6210 LC-MS TOF or Agilent Q-TOF 6520 LC-MS (Appendix A).

**General procedure 1: synthesis of compounds 3–4.** A mixture of methyltriphenylphosphonium bromide (2.79 g, 7.8 mmol) and sodium amide (0.30 g, 7.8 mmol) in anhydrous ether (20 mL) was stirred at 25 °C under argon protection for 1 h. The mixture was cooled to −10 °C, and **1** (0.59 g, 3.9 mmol) was added. The mixture continued to be stirred at 25 °C for another 10 min. The resulting precipitate was filtered off, washed with ethyl acetate, and concentrated in vacuo. The residue was purified through column chromatography on silica gel (petroleum ether/ethyl acetate = 10:3) to afford **3** (0.19 g, 32.9%). Compound **4** (0.21 g, 32.9%) was prepared according to the same procedure as **2** (0.65 g, 3.9 mmol).

**General procedure 2: synthesis of compounds 6 and 11.** To a solution of substituted styrene (1.3 mmol, 1.2 eq), tetrabutylammonium bromide (0.58 g, 1.8 mmol), potassium acetate (0.19 g, 1.9 mmol), and palladium acetate (0.02 g, 0.1 mmol) in DMF (25 mL) was added, along with substituted iodobenzene (1.1 mmol, 1 eq). The mixture was stirred at 80 °C for 5 h under argon. The reaction solution was extracted using ethyl acetate. The organic layer was combined, washed with saturated NaCl solution, and concentrated in vacuo to afford **6** (or **11**).

**Substituted (*E*)-1-nitro-4-styrylbenzene (6).** Compound **6** was prepared according to general procedure 2 from **5** (0.28 g, 1.1 mmol) and **3** (0.19 g, 1.3 mmol).

**General procedure 3: synthesis of compounds 7 and 28.** A well stirred mixture of compounds with a nitro group (10 mmol, 1 eq) and stannous chloride (40.0 mmol, 4 eq) in EtOH (40 mL) was heated to reflux for 2 h. After the reaction was completed, the solvent was removed, and the residue was stirred with NaOH (50 mL, 1 M) for another 0.5 h. The aqueous solution was extracted with ethyl acetate. The organic layer was combined, washed with saturated NaCl solution and then concentrated in vacuo. Purification through column chromatography on silica gel (dichloromethane = 100%) afforded products with an amino group.

**Substituted (*E*)-4-styrylaniline (7).** Compound **7** was prepared according to general procedure 3 from **6** (2.69 g, 10.0 mmol).

General procedure 4: synthesis of compounds **8**, **9**, **12**, **18**, **23**, **24**, **29**, **32**, **39**, and **42**.

**General procedure 4(a):** Acyl chloride (3.6 mmol, 3 eq) and pyridine (0.28 mL, 3.6 mmol) were added dropwise to a solution of compounds with an amino (1.2 mmol, 1 eq) and DMAP (0.02 g, 0.1 mmol) in CH_2_Cl_2_ (20 mL). The mixture was stirred at 25 °C from 1 h to 14 d. The mixture was poured into saturated NaHCO_3_ solution, the organic layer was separated, and the solvent was removed in vacuo. The residue was purified through column chromatography on silica gel to afford pure products.

**General procedure 4(b):** Pyridine (0.19 mL, 2.4 mmol) and anhydride (2.3 mmol, 3 eq) were added dropwise to a solution of compounds with an amino (0.8 mmol, 1 eq) and DMAP (0.02 g, 0.1 mmol) in CH_2_Cl_2_ (20 mL) below 0 °C. The mixture was stirred at 0 °C from 0.5 h to 1.5 h. The reaction solution was poured into saturated NaHCO_3_ solution, the organic layer was separated, and the solvent was removed in vacuo. The residue was purified through column chromatography on silica gel to afford pure target products.

**General procedure 4(c):** Pyridine (0.34 mL, 4.2 mmol) was added dropwise to a solution of compounds with an amino (1.0 mmol, 1 eq), DMAP (0.32 g, 2.6 mmol), DCC (0.54 g, 2.6 mmol), and carboxylic acid (2.6 mmol, 2.6 eq) in CH_2_Cl_2_ (20 mL) below 0 °C. The mixture was stirred at 0 °C for 1 h, then at 25 °C for 3 h. The reaction was filtered, then the filtrate was concentrated in vacuo to obtain compounds which were then purified through column chromatography. 

**(*E*)-3-(4-(2-(benzo[*d*]**[1,3]**dioxol-5-yl)vinyl)phenyl)-1,1-dimethylurea (8c).** Compound **8c** was prepared according to general procedure 4(a) from *N*, *N*-dimethylcarbamoyl chloride (0.32 mL, 3.6 mmol) and **7** (R^1^ R^2^ = OCH_2_O, R^3^ = H) (0.28 g, 1.2 mmol) for 1 h. Purification through column chromatography (dichloromethane/methanol = 10:0.1) on silica gel afforded pure **8c** (0.26 g, 68.9%) as a product. HRMS (ESI) (M+H)^+^ *m*/*z* 311.1397, calcd for C_18_H_19_N_2_O_3_ 311.1390. ^1^H NMR (Acetone-*d*_6_, 500 MHz) δ: 7.73 (s, 1H), 7.56–7.54 (m, 2H), 7.43–7.41 (m, 2H), 7.15 (d, *J* = 2.0 Hz, 1H), 7.03–6.98 (m, 3H), 6.82 (d, *J* = 8.5 Hz, 1H), 6.00 (s, 2H), 3.00 (s, 6H). ^13^C NMR (Acetone-*d*_6_, 125 MHz) δ: 156.4, 149.2, 148.0, 141.3, 133.4, 132.2, 127.7, 127.3, 127.0, 122.0, 120.3, 120.2, 109.1, 106.0, 102.1, 36.6.

**(*E*)-*N*-(4-(2-(benzo[*d*]**[1,3]**dioxol-5-yl)vinyl)phenyl)cyclobutanecarboxamide (8d).** Compound **8d** was prepared according to general procedure 4(c) from cyclobutanecarboxylic acid (0.83 g, 8.3 mmol) and **7** (R^1^ R^2^ = OCH_2_O, R^3^ = H) (0.76 g, 3.2 mmol) for 3 h. Purification through column chromatography (dichloromethane = 100%, petroleum ether/ethyl acetate = 10:3) on silica gel afforded pure **8d** (0.22 g, 21.2%) as a product. HRMS (ESI) (M+H)^+^ *m*/*z* 322.1445, calcd for C_20_H_20_NO_3_ 322.1438. ^1^H NMR (Acetone-*d*_6_, 500 MHz) δ: 8.91 (s, 1H), 7.67 (d, *J* = 8.5 Hz, 2H), 7.49–7.47 (m, 2H), 7.16 (d, *J* = 1.5 Hz, 1H), 7.06–7.00 (m, 3H), 6.83 (d, *J* = 8.0 Hz, 1H), 6.00 (s, 2H), 3.28–3.25 (m, 1H), 2.94–2.32 (m, 2H), 2.16–2.14 (m, 2H), 2.09–2.04 (m, 1H), 1.98–1.96 (m, 1H). ^13^C NMR (Acetone-*d*_6_, 125 MHz) δ: 173.6, 149.2, 148.2, 139.9, 133.5, 133.3, 127.8, 127.5, 127.4, 122.2, 120.2, 120.1, 109.1, 106.1, 102.1, 41.4, 25.7, 18.7.

**(*E*)-*N*-(4-(2-(benzo[*d*]**[1,3]**dioxol-5-yl)vinyl)phenyl)-1-cyanocyclopropane-1-carboxamide (8e).** Compound **8e** was prepared according to general procedure 4(c) from 1-cyanocyclopropylformic acid (0.29 g, 2.6 mmol) and **7** (R^1^ R^2^ = OCH_2_O, R^3^ = H) (0.24 g, 1.0 mmol) for 3 h. Purification through column chromatography (dichloromethane = 100%) on silica gel afforded pure **8e** (0.25 g, 73.5%) as a product. HRMS (ESI) (M+H)^+^ *m*/*z* 333.1235, calcd for C_20_H_17_N_2_O_3_ 333.1234. ^1^H NMR (DMSO-*d*_6_, 400 MHz) δ: 10.06 (s, 1H), 7.64 (d, *J* = 1.5 Hz, 2H), 7.50 (d, *J* = 8.5 Hz, 2H), 7.23 (s, 1H), 7.09 (d, *J* = 16.5 Hz, 1H), 7.04 (d, *J* = 16.5 Hz, 1H), 6.99 (d, *J* = 8.0 Hz, 1H), 6.87 (dd, *J* = 8.0 Hz, 2.5 Hz, 1H), 6.02 (s, 2H), 1.64–1.71 (m, 4H). ^13^C NMR (DMSO-*d*_6_, 100 MHz) δ: 163.7, 147.8, 146.9, 137.3, 133.3, 131.7, 127.4, 126.4, 126.1, 121.4, 121.1, 120.0, 108.3, 105.3, 101.1, 17.0, 14.8.

**(*E*)-*N*-(4-(2-(benzo[*d*]**[1,3]**dioxol-5-yl)vinyl)phenyl)-2-phenylacetamide (8g).** Compound **8g** was prepared according to general procedure 4(a) from phenylacetyl chloride (0.46 mL, 3.5 mmol) and **7** (R^1^ R^2^ = OCH_2_O, R^3^ = H) (0.28 g, 1.2 mmol) for 1 h. Purification through column chromatography (dichloromethane/methanol = 10:0.1) on silica gel afforded pure **8g** (0.28 g, 66.6%) as a product. HRMS (ESI) (M+H)^+^ *m*/*z* 358.1445, calcd for C_23_H_20_NO_3_ 358.1438. ^1^H NMR (Acetone-*d*_6_, 500 MHz) δ: 9.29 (s, 1H), 7.64–7.66 (m, 2H), 7.47–7.49 (m, 2H), 7.38–7.39 (m, 2H), 7.30–7.33 (m, 2H), 7.23–7.26 (m, 1H), 7.16 (d, *J* =1.5 Hz, 1H), 7.08 (d, *J* = 16.5 Hz, 1H), 7.04 (d, *J* = 16.5 Hz, 1H), 7.00 (dd, *J* = 8.0 Hz, 1.5 Hz, 1H), 6.82 (d, *J* = 8.0 Hz, 1H), 6.00 (s, 2H), 3.70 (s, 2H). ^13^C NMR (Acetone-*d*_6_, 125 MHz) δ: 169.7, 149.2, 148.2, 139.7, 136.9, 133.2, 130.1, 129.2, 128.0, 127.6, 127.5, 127.4, 122.2, 120.2, 120.2, 109.1, 106.1, 102.1, 44.8.

**(*E*)-*N*-(4-(2-(benzo[*d*]**[1,3]**dioxol-5-yl)vinyl)phenyl)picolinamide (8h).** Compound **8h** was prepared according to general procedure 4(c) from 2-picolinic acid (0.32 g, 2.6 mmol) and **7** (R^1^ R^2^ = OCH_2_O, R^3^ = H) (0.24 g, 1.0 mmol) for 1 h. Purification through column chromatography (dichloromethane = 100%, petroleum ether/ethyl acetate = 10:3) on silica gel afforded pure **8h** (0.16 g, 45.7%) as a product. HRMS (ESI) (M+H)^+^ *m*/*z* 345.1240, calcd for C_21_H_17_N_2_O_3_ 345.1234. ^1^H NMR (DMSO-*d*_6_, 500 MHz) δ: 10.55 (s, 1H), 8.64 (d, *J* = 1.5 Hz, 1H), 8.06 (d, *J* = 7.5 Hz, 1H), 7.96–7.99 (m, 1H), 7.82 (d, *J* = 8.5 Hz, 2H), 7.57–7.59 (m, 1H), 7.45 (d, *J* = 8.5 Hz, 2H), 7.15 (s, 1H), 7.03 (d, *J* = 16.0 Hz, 1H), 6.98 (d, *J* = 17.0 Hz, 1H), 6.91 (dd, *J* = 8.0 Hz, 1.0 Hz, 1H), 6.81 (d, *J* = 7.5 Hz, 1H), 5.93 (s, 2H). ^13^C NMR (DMSO-*d*_6_, 125 MHz) δ: 162.3, 149.8, 148.4, 147.8, 146.8, 138.1, 137.5, 133.0, 131.8, 127.2, 126.9, 126.6, 126.2, 122.3, 121.4, 120.3, 108.4, 105.2, 101.0.

**(*E*)-1-(4-(2-(benzo[*d*]**[1,3]**dioxol-5-yl)vinyl)phenyl)-*N*,*N*-dimethylmethanesulfonamide (8j).** Compound **8j** was prepared according to general procedure 4(a) from dimethylsulfamoyl chloride (0.27 mL, 2.5 mmol) and **7** (R^1^ R^2^ = OCH_2_O, R^3^ = H) (0.20 g, 0.8 mmol) for 14 d. Purification through column chromatography (dichloromethane/methanol = 10:0.1) on silica gel afforded pure **8j** (0.02 g, 6.9%) as a product. HRMS (ESI) (M+H)^+^ *m*/*z* 347.1055, calcd for C_17_H_19_N_2_O_4_S 347.1060. ^1^H NMR (Acetone-*d*_6_, 500 MHz) δ: 8.77 (s, 1H), 7.51–7.54 (m, 2H), 7.33–7.35 (m, 2H), 7.18 (d, *J* = 1.5 Hz, 1H), 7.11 (d, *J* = 16.5 Hz, 1H), 7.06 (d, *J* = 16.5 Hz, 1H), 7.01 (dd, *J* = 8.0 Hz, 2.0 Hz, 1H), 6.84 (d, *J* = 8.0 Hz, 1H), 6.01 (s, 2H), 2.79 (s, 6H). ^13^C NMR (Acetone-*d*_6_, 125 MHz) δ: 149.2, 148.2, 138.9, 134.2, 133.1, 128.3, 127.9, 127.1, 122.3, 120.9, 109.1, 106.1, 102.1, 38.4.

**(*E*)-*N*-(4-(2-(benzo[*d*]**[1,3]**dioxol-5-yl)vinyl)phenyl)cyclopropanesulfonamide (8k).** Compound **8k** was prepared according to general procedure 4(a) from cyclopropylsulfonyl chloride (0.25 mL, 2.5 mmol) and **7** (R^1^ R^2^ = OCH_2_O, R^3^ = H) (0.20 g, 0.8 mmol) for 6 d. Purification through column chromatography (dichloromethane = 100%) on silica gel afforded pure **8k** (0.24 g, 84.9%) as a product. HRMS (ESI) (M+H)^+^ *m*/*z* 344.0959, calcd for C_18_H_18_NO_4_S 344.0951. ^1^H NMR (DMSO-*d*_6_, 500 MHz) δ: 9.74 (s, 1H), 7.50 (d, *J* = 8.5 Hz, 2H), 7.24 (dd, *J* = 8.0 Hz, 1.5 Hz, 1H), 7.23 (d, *J* = 8.5 Hz, 2H), 7.08 (d, *J* = 16.5 Hz, 1H), 7.05 (d, *J* = 16.5 Hz, 1H), 7.00 (dd, *J* = 8.0 Hz, 1.5 Hz, 1H), 6.89 (d, *J* = 8.0 Hz, 1H), 6.02 (s, 2H), 2.60–2.64 (m, 1H), 0.92–0.94 (m, 4H). ^13^C NMR (DMSO-*d*_6_, 125 MHz) δ: 147.8, 146.8, 137.4, 133.0, 131.7, 127.3, 127.0, 126.0, 121.4, 120.4, 108.3, 105.2, 101.0, 29.6, 4.9.

**(*E*)-*N*-(4-(2-(benzo[*d*]**[1,3]**dioxol-5-yl)vinyl)phenyl)benzenesulfonamide (8l).** Compound **8l** was prepared according to general procedure 4(a) from benzenesulfonyl chloride (0.46 mL, 3.6 mmol) and **7** (R^1^ R^2^ = OCH_2_O, R^3^ = H) (0.29 g, 1.2 mmol) for 1 h. Purification through column chromatography (dichloromethane = 100%) on silica gel afforded pure **8l** (0.20 g, 43.3%) as a product. HRMS (ESI) (M+H)^+^ *m*/*z* 380.0950, calcd for C_21_H_18_NO_4_S 380.0951. ^1^H NMR (DMSO-*d*_6_, 500 MHz) δ: 10.30 (s, 1H), 7.77 (d, *J* = 7.0 Hz, 2H), 7.52–7.76 (m, 3H), 7.40 (d, *J* = 8.5 Hz, 2H), 7.20 (s, 1H), 7.08 (d, *J* = 8.5 Hz, 2H), 7.02 (d, *J* = 16.0 Hz, 1H), 6.97 (d, *J* = 16.0 Hz, 1H), 6.96 (d, *J* = 8.0 Hz, 1H), 6.88 (d, *J* = 8.0 Hz, 1H), 6.01 (s, 2H). ^13^C NMR (Acetone-*d*_6_, 125 MHz) δ: 149.2, 148.2, 140.9, 137.7, 135.1, 133.6, 132.9, 129.9, 128.7, 127.9, 126.8, 122.3, 122.0, 109.1, 106.1, 102.1.

**(*E*)-*N*-(4-(3,5-dimethoxystyryl)phenyl)isobutyramide (9b).** Compound **9b** was prepared according to general procedure 4(b) from isobutyric anhydride (0.38 mL, 2.3 mmol) and **7** (R^1^, R^3^ = OCH_3_, R^2^ =H) (0.20 g, 0.8 mmol) for 1.5 h. Purification through column chromatography (dichloromethane/methanol = 10:0.1) on silica gel afforded pure **9b** (0.10 g, 41.4%) as a product. HRMS (ESI) (M+H)^+^ *m*/*z* 326.1746, calcd for C_20_H_24_NO_3_ 326.1751. ^1^H NMR (DMSO-*d*_6_, 500 MHz) δ: 9.90 (s, 1H), 7.65 (d, *J* = 8.5 Hz, 2H), 7.52 (d, *J* = 8.5 Hz, 2H), 7.21 (d, *J* = 16.5 Hz, 1H), 7.07 (d, *J* = 16.5 Hz, 1H), 6.76 (d, *J* = 2.0 Hz, 2H), 6.41 (t, *J* = 2.0 Hz, 1H), 3.78 (s, 6H), 2.58–2.64 (m, 1H), 1.12 (s, 3H), 1.11 (s, 3H). ^13^C NMR (DMSO-*d*_6_, 125 MHz) δ: 175.2, 160.6, 139.3, 139.0, 131.6, 128.6, 126.9, 126.9, 119.2, 104.2, 99.6, 55.1, 35.0, 19.5.

**(*E*)-4-(3,5-dimethoxystyryl)-2-fluoroaniline (11).** Compound **11** (1.89 g, 49.2%) was prepared according to general procedure 2 from **4** (1.60 g, 9.7 mmol) and **10** (2.00 g, 8.4 mmol).

**(*E*)-*N*-(4-(3,5-dimethoxystyryl)-2-fluorophenyl)benzamide (12f).** Compound **12f** was prepared according to general procedure 4(b) from benzoic anhydride (0.68 g, 3.0 mmol) and **11** (0.28 g, 1.0 mmol) for 1 h. Purification through column chromatography (dichloromethane = 100%) on silica gel afforded pure **12f** (0.15 g, 40.4%) as a product. HRMS (ESI) (M+H)^+^ *m*/*z* 378.1503, calcd for C_23_H_21_FNO_3_ 378.1500. ^1^H NMR (DMSO-*d*_6_, 500 MHz) δ: 10.17 (s, 1H), 8.03 (d, *J* = 7.5 Hz, 2H), 7.69 (t, *J* = 8.0 Hz, 1H), 7.53–7.63 (m, 4H), 7.48 (dd, *J* = 8.0 Hz, 1.5 Hz, 1H), 7.31 (d, *J* = 16.5 Hz, 1H), 7.26 (d, *J* = 16.5 Hz, 1H), 6.82 (d, *J* = 2.0 Hz, 2H), 6.47 (t, *J* = 2.0 Hz, 1H), 3.80 (s, 6H). ^13^C NMR (DMSO-*d*_6_, 125 MHz) δ: 165.4, 160.7, 156.7, 154.8, 138.8, 136.1, 134.0, 131.8, 129.4, 128.4, 127.8, 127.4, 126.8, 125.1, 122.6,113.1, 104.6, 100.1, 55.2.

**General procedure 5: synthesis of compounds 15 and 20.** To a solution of LiOH•H_2_O (12.08 g, 288.0 mmol, 16 eq) in MeOH (15 mL) was added substituted acetophenone (18.0 mmol, 1 eq). The mixture was stirred at 25 °C for 1 h, then substituted benzaldehyde (18.0 mmol, 1 eq) was added. The mixture was stirred for another 3 h and then concentrated under reduced pressure. To the residue was added distilled water (30 mL), and the aqueous solution was acidified with hydrochloric acid (10%) to pH 3–4. The resulting precipitate was collected through filtration and washed with cold MeOH to afford pure products.

**(*E*)-1-(benzo[*d*]**[1,3]**dioxol-5-yl)-3-(4-nitrophenyl)prop-2-en-1-one (15).** Compound **15** (5.25 g, 97.9%) was prepared according to general procedure 5 from **13** (2.95 g, 18.0 mmol) and **14** (2.72 g, 18.0 mmol).

**General procedure 6: synthesis of compounds 16 and 21.** To a solution of anhydrous nickel chloride (2.63 g, 20.0 mmol) and chalcone (10.0 mmol) in EtOH (75 mL) was added NaBH_4_ (2.3 g, 60.0 mmol). The mixture was stirred at 25 °C for 5 min under argon. When the reaction was completed, the mixture was filtered, poured into distilled water (20 mL), then extracted with CH_2_Cl_2_. The organic layer was washed with saturated NaCl solution and concentrated in vacuo to afford the products. 

**3-(4-aminophenyl)-1-(benzo[*d*]**[1,3]**dioxol-5-yl)propan-1-ol (16).** Compound **16** was prepared according to general procedure 6 from **15** (2.97 g, 10.0 mmol).

**General procedure 7: synthesis of compounds 17 and 22.** A well-stirred mixture of *p*-toluenesulfonic acid (0.35 g) and **16** or **21** in toluene (50 mL) was heated to reflux for 3 h, then cooled to 25 °C. The mixture was poured into a saturated NaHCO_3_ solution to stir for 0.5 h and then concentrated in vacuo. The residue was purified through column chromatography on silica gel to afford **17** or **22**.

**(*E*)-4-(3-(benzo[*d*][1,3]dioxol-5-yl)allyl)aniline (17).** Compound **17** was prepared according to general procedure 7 from **16**. Purification through column chromatography (dichloromethane = 100%) on silica gel afforded pure **17** (0.26 g, 14.3%) as a product. 

**(*E*)-*N*-(4-(3-(benzo[*d*]**[1,3]**dioxol-5-yl)allyl)phenyl)isobutyramide (18b).** Compound **18b** was prepared according to general procedure 4(b) from isobutyric anhydride (0.33 mL, 2.0 mmol) and **17** (0.17 g, 0.7 mmol). Purification through column chromatography (dichloromethane/methanol = 10:0.1) on silica gel afforded pure **18b** (0.12 g, 57.8%) as a product. HRMS (ESI) (M+H)^+^ *m*/*z* 324.1586, calcd for C_20_H_22_NO_3_ 324.1594. ^1^H NMR (DMSO-*d*_6_, 500 MHz) δ: 9.75 (s, 1H), 7.54 (d, *J* = 8.5 Hz, 2H), 7.14 (d, *J* = 8.5 Hz, 2H), 7.06 (s, 1H), 6.80–6.84 (m, 2H), 6.35 (d, *J* = 16.0 Hz, 1H), 6.26 (dt, *J* = 16.0 Hz, 7.0 Hz, 1H), 5.98 (s, 2H), 3.41 (d, *J* = 7.0 Hz, 2H), 2.54–2.60 (m, 1H), 1.10 (s, 3H), 1.08 (s, 3H). ^13^C NMR (DMSO-*d*_6_, 125 MHz) δ: 175.0, 147.7, 146.4, 137.5, 134.6, 131.6, 130.0, 128.5, 127.7, 120.5, 119.3, 108.2, 105.2, 100.8, 37.9, 34.8, 19.5.

**Substituted (*E*)-1-(4-nitrophenyl)-3-phenylprop-2-en-1-one (20).** Compound **20** (99.2%) was prepared according to general procedure 5 from **19** (3.00 g, 18.0 mmol) and **1** (2.70 g, 18.0 mmol) or **2** (2.99 g, 18.0 mmol).

**Substituted 1-(4-aminophenyl)-3-phenylpropan-1-ol (21).** Compound **21** was prepared according to general procedure 6 from **20**.

**Substituted (*E*)-4-(3-phenylprop-1-en-1-yl)aniline (22).** Compound **22** was prepared according to general procedure 7 from **21**. Purification through column chromatography (dichloromethane = 100%) on silica gel afforded pure **22** as a product.

**(*E*)-*N*-(4-(3-(benzo[*d*]**[1,3]**dioxol-5-yl)prop-1-en-1-yl)phenyl)isobutyramide (23b).** Compound **23b** was prepared according to general procedure 4(b) from isobutyric anhydride (0.33 mL, 2.0 mmol) and **22** (R^1^ R^2^ = OCH_2_O, R^3^ = H) (0.17 g, 0.7 mmol) for 0.5 h. Purification through column chromatography (dichloromethane = 100%) on silica gel afforded pure **23b** (0.10 g, 47.5%) as a product. HRMS (ESI) (M+H)^+^ *m*/*z* 324.1586, calcd for C_20_H_22_NO_3_ 324.1594. ^1^H NMR (DMSO-*d*_6_, 500 MHz) δ: 9.81 (s, 1H), 7.56 (d, *J* = 8.0 Hz, 2H), 7.31 (d, *J* = 8.0 Hz, 2H), 6.80–6.83 (m, 2H), 6.69 (d, *J* = 8.0 Hz, 1H), 6.37 (d, *J* = 15.5 Hz, 1H), 6.27 (dt, *J* = 15.5 Hz, 6.5 Hz, 1H), 5.96 (s, 2H), 3.40 (d, *J* = 6.5 Hz, 2H), 2.55–2.60 (m, 1H), 1.10 (s, 3H), 1.08 (s, 3H). ^13^C NMR (DMSO-*d*_6_, 125 MHz) δ: 175.0, 147.3, 145.4, 138.4, 134.0, 131.8, 129.8, 128.0, 126.2, 121.2, 119.1, 108.9, 108.1, 100.6, 38.2, 34.9, 19.5.

**(*E*)-*N*-(4-(3-(benzo[*d*]**[1,3]**dioxol-5-yl)prop-1-en-1-yl)phenyl)methanesulfonamide (23i).** Compound **23i** was prepared according to general procedure 4(a) from methanesulfonyl chloride (0.23 mL, 2.9 mmol) and **22** (R^1^ R^2^ = OCH_2_O, R^3^ = H) (0.25 g, 1.0 mmol) for 0.5 h. Purification through column chromatography (dichloromethane/methanol = 10:0.1) on silica gel afforded pure **23i** (0.11 g, 32.7%) as a product. HRMS (ESI) (M+NH_4_)^+^ *m*/*z* 349.1210, calcd for C_17_H_21_N_2_O_4_S 349.1217. ^1^H NMR (DMSO-*d*_6_, 500 MHz) δ: 9.72 (s, 1H), 7.37 (d, *J* = 8.5 Hz, 2H), 7.15 (d, *J* = 8.5 Hz, 2H), 6.81–6.84 (m, 2H), 6.71 (dd, *J* = 7.5 Hz, 0.5 Hz, 1H), 6.40 (d, *J* = 16.0 Hz, 1H), 6.31 (dt, *J* = 16.0 Hz, 7.0 Hz, 1H), 5.97 (s, 2H), 3.42 (d, *J* = 7.0 Hz, 2H), 2.96 (s, 3H). ^13^C NMR (DMSO-*d*_6_, 125 MHz) δ: 147.3, 145.5, 137.2, 133.9, 132.9, 129.6, 128.8, 126.9, 121.2, 119.9, 108.9, 108.2, 100.7, 39.1, 38.2.

**(*E*)-*N*-(4-(3-(3,5-dimethoxyphenyl)prop-1-en-1-yl)phenyl)isobutyramide (24b).** Compound **24b** was prepared according to general procedure 4(b) from isobutyric anhydride (0.76 mL, 4.7 mmol) and **22** (R^1^, R^3^ = OCH_3_, R^2^ = H) (0.42 g, 1.6 mmol) for 1.5 h. Purification through column chromatography (dichloromethane = 100%) on silica gel afforded pure **24b** (0.29 g, 54.8%) as a product. HRMS (ESI) (M+H)^+^ *m*/*z* 340.1926, calcd for C_21_H_26_NO_3_ 340.1907. ^1^H NMR (DMSO-*d*_6_, 500 MHz) δ: 9.83 (s, 1H), 7.59 (d, *J* = 8.5 Hz, 2H), 7.33 (d, *J* = 8.5 Hz, 2H), 6.42 (d, *J* = 2.0 Hz, 2H), 6.38 (s, 1H), 6.35 (t, *J* = 2.0 Hz, 1H), 6.30 (dt, *J* = 16.0 Hz, 6.5 Hz, 1H), 3.72 (s, 6H), 3.41–3.44 (m, 2H), 2.56–2.64 (m, 1H), 1.11 (s, 3H), 1.10 (s, 3H). ^13^C NMR (DMSO-*d*_6_, 125 MHz) δ: 175.2, 160.6, 142.6, 138.5, 131.9, 130.2, 127.5, 126.3, 119.2, 106.5, 97.9, 55.0, 38.9, 35.0, 19.5.

**General procedure 8: synthesis of compounds 27 and 30.** To a mixture of substituted phenylacetonitrile (5.5 mmol, 1.1 eq) and **1** (0.75 g, 5.0 mmol) or **14** (0.76 g, 5.0 mmol) in anhydrous methanol (23 mL), sodium methoxide (4 mL, 5 mmol/L) was added dropwise. Under argon protection, the mixture was stirred at reflux temperature for 3 h and then cooled to 25 °C. The mixture was poured into cold distilled water. The resulting precipitate was collected through filtration and washed with cold distilled methanol to afford **27** or **30**.

**(*Z*)-3-(benzo[*d*]**[1,3]**dioxol-5-yl)-2-(4-nitrophenyl)acrylonitrile (27).** Compound **27** (0.40 g, 27.2%) was prepared according to general procedure 8 from **1** (0.75 g, 5.0 mmol) and **25** (0.89 g, 5.5 mmol). 

**(*Z*)-2-(4-aminophenyl)-3-(benzo[*d*]**[1,3]**dioxol-5-yl)acrylonitrile (28).** Compound **28** (0.13 g, 37.6%) was prepared according to general procedure 3 from **27** (0.40 g, 1.35 mmol).

**(*Z*)-*N*-(4-(2-(benzo[*d*]**[1,3]**dioxol-5-yl)-1-cyanovinyl)phenyl)acetamide (29a).** Compound **29a** was prepared according to general procedure 4(b) from acetic anhydride (0.54 mL, 5.7 mmol) and **28** (0.50 g, 1.9 mmol) for 1.5 h. Purification through column chromatography (dichloromethane/methanol = 10:0.1) on silica gel afforded pure **29a (**0.36 g, 61.9%) as a product. HRMS (ESI) (M+H)^+^ *m*/*z* 307.1070, calcd for C_18_H_15_N_2_O_3_ 307.1077. ^1^H NMR (DMSO-*d*_6_, 400 MHz) δ: 10.15 (s, 1H), 7.83 (s, 1H), 7.73–7.67 (m, 2H), 7.67–7.62 (m, 2H), 7.57 (d, *J* = 1.8 Hz, 1H), 7.43 (dd, *J* = 1.8, 8.5 Hz, 1H), 7.09 (d, *J* = 8.2 Hz, 1H), 6.14 (s, 2H), 2.07 (s, 3H). ^13^C NMR (DMSO-*d*_6_, 101 MHz) δ: 169.02, 149.63, 148.25, 141.23, 140.47, 128.87, 128.50, 126.52, 126.01, 119.64, 118.76, 109.26, 107.97, 107.84, 102.34, 24.54.

**(*Z*)-*N*-(4-(2-(benzo[*d*]**[1,3]**dioxol-5-yl)-1-cyanovinyl)phenyl)isobutyramide (29b).** Compound **29b** was prepared according to general procedure 4(b) from isobutyric anhydride (0.64 mL, 3.9 mmol) and **28** (0.34 g, 1.3 mmol) for 1 h. Purification through column chromatography (dichloromethane = 100%) on silica gel afforded pure **29b** (0.12 g, 28.8%) as a product. HRMS (ESI) (M+H)^+^ *m*/*z* 335.1397, calcd for C_20_H_19_N_2_O_3_ 335.1390. ^1^H NMR (DMSO-*d*_6_, 500 MHz) δ: 10.02 (s, 1H), 7.82 (s, 1H), 7.73 (d, *J* = 8.5 Hz, 2H), 7.64 (d, *J* = 8.5 Hz, 2H), 7.56 (s, 1H), 7.42 (d, *J* = 8.0 Hz, 1H), 7.06 (d, *J* = 8.0 Hz, 1H), 6.12 (s, 2H), 2.58–2.64 (m, 1H), 1.12 (s, 3H), 1.10 (s, 3H). ^13^C NMR (DMSO-*d*_6_, 125 MHz) δ: 175.5, 149.1, 147.8, 140.6, 140.1, 128.3, 128.0, 125.9, 125.5, 119.3, 118.3, 108.7, 107.4, 107.3, 101.8, 35.0, 19.4.

**(*Z*)-2-(benzo[*d*]**[1,3]**dioxol-5-yl)-3-(4-nitrophenyl)acrylonitrile (30).** Compound **30** (0.52 g, 17.8%) was prepared according to general procedure 8 from **14** (1.662 g, 11 mmol) and **26** (1.612 g, 10 mmol).

**(*Z*)-3-(4-aminophenyl)-2-(benzo[*d*]**[1,3]**dioxol-5-yl)acrylonitrile (31).** To a suspension of iron powder (0.40 g, 7.1 mmol) and ammonium chloride (0.24 g, 4.4 mmol) in water (20 mL), **30** in methanol (20 mL) was added dropwise. The mixture was stirred at 80 °C for 2 h. After the reaction was completed, the mixture was extracted with ethyl acetate (50 mL) and alkalified with NaOH (2M) to adjust the pH to 9–10. The organic layer was washed sequentially with saturated NH_4_Cl, saturated NaHCO_3_, water, and saturated NaCl solution. Recrystallization from ethyl acetate and petroleum ether afforded compound **31** (0.39 g, 82.2%).

**(*Z*)-*N*-(4-(2-(benzo[*d*]**[1,3]**dioxol-5-yl)-2-cyanovinyl)phenyl)acetamide (32a).** Compound **32a** was prepared according to general procedure 4(b) from acetic anhydride (0.30 mL, 3.1 mmol) and **31** (0.28 g, 1.0 mmol) for 1.5 h. Recrystallization from dichloromethane and petroleum ether afforded pure **32a** (0.10 g, 31.7%) as a product. HRMS (ESI) (M+H)^+^ *m*/*z* 307.1079, calcd for C_18_H_15_N_2_O_3_ 307.1077. ^1^H NMR (DMSO-*d*_6_, 400 MHz) δ: 10.23 (s, 1H), 7.86 (d, *J* = 8.6 Hz, 2H), 7.83 (s, 1H), 7.72 (d, *J* = 8.5 Hz, 2H), 7.39 (d, *J* = 2.0 Hz, 1H), 7.19 (dd, *J* = 2.0, 8.1 Hz, 1H), 7.03 (d, *J* = 8.2 Hz, 1H), 6.11 (s, 2H), 2.09 (s, 3H). ^13^C NMR (DMSO-*d*_6_, 101 MHz) δ: 169.22, 148.77, 148.51, 141.66, 141.22, 130.42, 128.82, 128.77, 120.78, 119.24, 118.80, 109.07, 108.07, 105.71, 102.19, 24.60.

**(*Z*)-*N*-(4-(2-(benzo[*d*]**[1,3]**dioxol-5-yl)-2-cyanovinyl)phenyl)isobutyramide (32b).** Compound **32b** was prepared according to general procedure 4(b) from isobutyric anhydride (1.74 mL, 10.7 mmol) and **31** (0.95 g, 3.6 mmol) for 1 h. Purification through column chromatography (dichloromethane = 100%) on silica gel afforded pure **32b** (0.325 g, 27.2%) as a product. HRMS (ESI) (M+H)^+^ *m*/*z* 335.1390, calcd for C_20_H_19_N_2_O_3_ 335.1390. ^1^H NMR (DMSO-*d*_6_, 400 MHz) δ: 10.14 (s, 1H), 7.99–7.81 (m, 3H), 7.76 (d, *J* = 8.8 Hz, 2H), 7.39 (d, *J* = 2.0 Hz, 1H), 7.19 (dd, *J* = 2.0, 8.2 Hz, 1H), 7.03 (d, *J* = 8.2 Hz, 1H), 6.11 (s, 2H), 2.70–2.56 (m, 1H), 1.12 (d, *J* = 6.8 Hz, 6H). ^13^C NMR (DMSO-*d*_6_, 101 MHz) δ: 176.12, 148.76, 148.50, 141.82, 141.24, 130.38, 128.78, 120.77, 119.39, 118.81, 109.07, 108.05, 105.70, 102.19, 35.51, 19.90.

***N*-(4-(((benzo[*d*]**[1,3]**dioxol-5-ylmethyl)amino)methyl)phenyl)acetamide (35).** A mixture of **33** (0.20 g, 1.2 mmol), **34** (0.18 mL, 1.4 mmol), NaBH_4_ (0.06 g, 1.6 mmol), and molecular sieve 4A (5.00 g) in anhydrous CHCl_3_ (20 mL) was stirred at 25 °C for 10 h. The mixture was quenched with distilled water (20 mL), then extracted with ethyl acetate. The organic layer was washed with saturated NaCl solution, then concentrated in vacuo. The residue was purified through column chromatography on silica gel (dichloromethane/methanol/ammonia = 10:0.3:0.03) to afford **35** (0.21 g, 58.7%) as a product.

***N*-(4-(((benzo[*d*]**[1,3]**dioxol-5-ylmethyl)(methyl)amino)methyl)phenyl)acetamide (36a).** Aqueous formaldehyde solution (0.14 mL, 37%) and formic acid (0.13 mL, 98%) were added dropwise to a solution of **35** (0.21 g, 0.7 mmol) in anhydrous CHCl_3_ (30 mL). The mixture was stirred at reflux temperature for 3 h. The mixture was quenched by distilled water (20 mL), acidified through the addition of HCl (30%) to pH 5, then extracted with CHCl_3_ (20 mL). The organic layer was alkalified with NaOH solution to adjust the pH to 9, then concentrated in vacuo. The residue was purified through column chromatography on silica gel (dichloromethane/methanol/ammonia = 10:0.5:0.03) to afford **36a** (0.17 g, 44.3%) as a product. HRMS (ESI) (M+H)^+^ *m*/*z* 313.1542, calcd for C_18_H_21_N_2_O_3_ 313.1547. ^1^H NMR (CDCl_3_, 500 MHz) δ: 8.35 (s, 1H), 7.47 (d, *J* = 8.0 Hz, 2H), 7.24 (d, *J* = 8.0 Hz, 2H), 6.86 (d, *J* = 1.0 Hz, 1H), 6.71–6.75 (m, 2H), 5.89 (s, 2H), 3.42 (s, 2H), 3.38 (s, 2H), 2.11 (s, 6H). ^13^C NMR (CDCl_3_, 125 MHz) δ: 168.9, 147.4, 146.3, 136.8, 134.9, 133.0, 129.2, 121.7, 119.9, 109.1, 107.6, 100.6, 61.2, 60.9, 41.8, 24.1.

**General procedure 9: synthesis of compounds 38 and 41.** To a solution of **3** (0.74 g, 5.0 mmol), P(*o*-MePh)_3_ (0.61 g, 2.0 mmol), triethylamine (0.21 mL, 1.5 mmol), and palladium acetate (0.22 g, 1.0 mmol) in CH_3_CN (20 mL) was added the derivative of the aromatic halogen (1 eq). Under argon protection, the mixture was stirred at 60 °C for 1 h, then warmed to 90 °C for 24 h. After the reaction was completed, the mixture was extracted with CH_2_Cl_2_ (50 mL), then washed with water and saturated NaCl solution. The organic layer was concentrated in vacuo. Recrystallization from ethyl acetate and petroleum ether afforded **38** or **41.**

**(*E*)-4-(2-(benzo[*d*]**[1,3]**dioxol-5-yl)vinyl)benzoic acid (38). 37** (1.24 g, 5 mmol) was used to yield the compound **38** (1.15 g, 86.0%) according to the general procedure 9.

**(*E*)-4-(2-(benzo[*d*]**[1,3]**dioxol-5-yl)vinyl)-*N*-isopropylbenzamide (39b).** To a solution of compound **38** in CH_2_Cl_2_ (40 mL), triethylamine (1.10 mL, 8.0 mmol) and pivaloyl chloride (0.98 mL, 8.0 mmol) were added dropwise. The mixture was stirred at 0 °C for 0.5 h, and isopropylamine (2.00 mL, 24 mmol) and DMAP (0.98 g, 8 mmol) were added. After the reaction was completed, the mixture was washed with saturated NH_4_Cl, saturated NaHCO_3_, distilled water, and saturated NaCl solution, then concentrated in vacuo. Recrystallization from ethyl acetate provided the compound **39b** (1.46 g, 59.1%). HRMS (ESI) (M+H)^+^ *m*/*z* 310.1442, calcd for C_19_H_20_NO_3_ 310.1438. ^1^H NMR (DMSO-*d*_6_, 500 MHz) δ: 8.19 (d, *J* = 7.8 Hz, 1H), 7.98–7.78 (m, 2H), 7.62 (d, *J* = 8.4 Hz, 2H), 7.38–7.23 (m, 2H), 7.16 (d, *J* = 16.3 Hz, 1H), 7.07 (dd, *J* = 1.7, 8.1 Hz, 1H), 6.94 (d, *J* = 8.0 Hz, 1H), 6.05 (s, 2H), 4.16–4.06 (m, 1H), 1.18 (d, *J* = 6.6 Hz, 6H). ^13^C NMR (DMSO-*d*_6_, 126 MHz) δ: 165.39, 148.39, 147.74, 140.29, 133.68, 131.82, 130.29, 128.14, 126.30, 126.27, 122.55, 108.92, 105.88, 101.64, 41.41, 22.84.

**(*E*)-(4-(2-(benzo[*d*]**[1,3]**dioxol-5-yl)vinyl)phenyl)methanamine (41).** Compound **40** (0.93 g, 5.0 mmol) was used to obtain **41** (0.99 g, 78.1%) according to general procedure 9.

**(*E*)-*N*-(4-(2-(benzo[*d*]**[1,3]**dioxol-5-yl)vinyl)benzyl)isobutyramide (42b).** Compound **42** was prepared according to general procedure 4(b) from isobutyric anhydride (0.24 mL, 1.5 mmol) and **41** (0.13 g, 0.5 mmol). Recrystallization from dichloromethane and petroleum ether afforded **42b** (0.14 g, 86.6%). HRMS (ESI) (M+H)^+^ *m*/*z* 324.1609, calcd for C_20_H_22_NO_3_ 324.1594. ^1^H NMR (DMSO-*d*_6_, 400 MHz) δ: 8.28 (t, *J* = 6.0 Hz, 1H), 7.51 (d, *J* = 7.8 Hz, 2H), 7.32–7.21 (m, 3H), 7.12 (d, *J* = 9.5 Hz, 2H), 7.03 (dd, *J* = 1.7, 8.1 Hz, 1H), 6.90 (dd, *J* = 1.3, 8.0 Hz, 1H), 6.04 (s, 2H), 4.28 (d, *J* = 6.0 Hz, 2H), 2.48–2.40 (m, 1H), 1.07 (d, *J* = 6.9 Hz, 6H). ^13^C NMR (DMSO-*d*_6_, 101 MHz) δ: 176.60, 148.36, 147.40, 139.45, 136.25, 132.11, 128.26, 127.89, 126.86, 126.64, 122.04, 108.84, 105.76, 101.55, 42.16, 34.54, 20.07.

### 3.2. Animal Experiments

Eight-week-old (or 10-week-old) SD male rats purchased from Peking University Health Science Center were acclimatized for one week at the Beijing Institute of Technology. All experimental protocols strictly adhered to the regulations governing the management of experimental animals and the animal ethics policy of the Beijing Institute of Technology. 

The compounds were dissolved in reagent-grade soybean oil containing 10% ethanol to reach a drug concentration of 2.8 mg/mL. The rats in the experimental group received intraperitoneal injections of the compounds at a dose of 4.0 mg/kg/d for 28 consecutive days. As a control, the vehicle group was administered injections of reagent-grade soybean oil containing 10% ethanol. BrdU (Sigma-Aldrich, St. louis, MA, USA) solution, prepared in physiological saline at a concentration of 10 mg/mL, was administered to all rats on day 29. Each rat received two injections of BrdU at a dosage of 50 mg/kg, with a 2 h interval between injections. The rats were euthanized 24 h (or 28 days) after the final BrdU injection.

Two or four rats per group were anesthetized using pentobarbital sodium (70 mg/kg) and subsequently perfused with physiological saline. The brain tissue was carefully dissected and fixed in 4% paraformaldehyde at 4 °C for one week. Next, the brain tissue underwent dehydration using a phosphate-buffered saline (PBS) solution containing 20% to 30% sucrose. Finally, the brain tissue was cryosectioned into slices with a thickness of 30 μm, and eight slices were randomly selected for further analysis.

The assessment of new cell proliferation was conducted using BrdU staining. To eliminate endogenous peroxides, sections were exposed to 1% H_2_O_2_ for 30 min at RT. Cellular denaturation was achieved by treating the sections with 2 M HCl at 37 °C for 1 h, followed by rinsing in 0.1 M borate buffer (pH 8.5). Tissue sections were subjected to blocking in a solution consisting of 5% goat serum and 0.5% Triton X-100 in PBS at RT for 1–2 h. The specimens were then incubated overnight at 4 °C with a mouse primary antibody specific to BrdU (diluted at 1:500; Millipore, Billerica, MA, USA). Subsequently, the DAB kit was employed for 2 h of incubation at RT.

The differentiation process of NPCs was visualized through the utilization of the NeuN/BrdU double staining technique. Tissue sections were subjected to an incubation period in a blocking buffer at RT for 1 to 2 h. The specimens were exposed to a rabbit *anti*-NeuN antibody (diluted at 1:500, Abcam, Cambridge, MA, UK) and a primary antibody, BrdU (diluted at 1:500), overnight at 4 °C. Afterwards, the sections were treated with anti-mouse 488 and anti-rabbit TRITC secondary antibodies for 2 h at RT. 

The data were presented as mean ± SEM (standard error of the mean) and visualized using GRAPHPAD PRISM 6 (Graphpad Software, La Jolla, CA, USA). Statistical analysis was conducted utilizing one-way analysis of variance (ANOVA) followed by a post hoc test. Subsequent to the statistical treatment, the observed discrepancies were determined to be statistically significant (*p* < 0.05).

### 3.3. Pharmacokinetic Study

The SD rats (approximately 200 g) were randomly assigned to eight groups, with each group comprising three rats. All experimental procedures were carried out in accordance with the ethical guidelines outlined by the Ethics Committee of the Beijing Institute of Technology. The administration medium consisted of a combination of 5% dimethyl sulfoxide (DMSO) and 95% soybean oil. The drug was administered intraperitoneally at a dose of 4 mg/kg. Prior to the experiment, the experimental animals were subjected to a 12 h fasting period with free access to water. Following drug administration, blood samples were collected from the rat’s orbital vein at predetermined time points. The collected blood was then placed in sodium heparin anticoagulation tubes. The rats were euthanized, and their brain tissues were carefully extracted and loaded into 5 mL EP tubes. The collection time points for plasma and brain tissue samples were as follows: 0 min, 15 min, 30 min, 1 h, 2 h, 4 h, 8 h, and 24 h. Within 2 h after collection, the blood samples were centrifuged at 6000 rpm for 10 min at a temperature of 4 °C. The resulting samples were then stored in a −20 °C refrigerator. 

Tissue samples were first weighed and homogenized to ensure consistent testing conditions. The determination of original compound concentrations in plasma and tissue homogenates was performed using LC-MS/MS. Specifically, 50 μL of plasma or tissue samples were mixed with 50 μL of propranolol and 100 μL of acetonitrile for protein precipitation. The resulting mixture was then centrifuged at 12,000 rpm for 5 min, and the supernatant was diluted and analyzed through LC-MS/MS. The liquid chromatography was conducted on a Thermo Scientific Q Exactive HF-X instrument (Thermo Fisher Scientific, Waltham, MA, USA), employing an ACQUITY UPLC BEH C18 Column with dimensions of 130 Å, 1.7 µm, 2.1 mm × 100 mm. Additionally, a pre-column equipped with an ACQUITY UPLC BEH C18 VanGuard Pre-Column (130 Å, 1.7 µm, 2.1 mm × 5 mm) was utilized. The liquid phase method employed gradient elution with mobile phases consisting of acetonitrile and a 0.1% formic acid aqueous solution. Mass spectrometry was conducted in positive ion mode. The quantification of drug concentration in plasma samples was performed using the internal standard method, and the obtained pharmacokinetic parameters were calculated using Phoenix WinNonlin 7.0 software.

### 3.4. CCK-8 Assay for Cytotoxicity

hCMEC/D3 cells were obtained from the Cell Resource Center, Peking Union Medical College (PCRC). All cells were acquired in 2023 and were maintained in culture for no more than 10 continuous passages. The cells were cultured in DMEM (11995065, Gibco, Grand Island, NY, USA) supplemented with 10% FBS (164210-50, Procell, Austin, TX, USA), 100 units of penicillin, and 100 mg/mL of streptomycin (PB180120, Procell).

hCMEC/D3 cells (5 × 10^3^) were seeded into 96-well plates in 90 μL of DMEM medium containing 10% fetal bovine serum per well. All test compounds were formulated as 10 mM stock solutions in DMSO (final concentration ≤ 0.5%). After overnight incubation, the test compound stock solution was diluted with DMEM medium containing 10% fetal bovine serum to every target concentration, and 10 μL of each was added to each well. Three parallel experiments were set up for each compound. After incubation for 48 h, 10 μL of CCK-8 reagent was added to each well in sequence according to the manufacturer’s instructions. After incubating in a 37 °C incubator for 1–2 h, the absorbance was measured at a wavelength of 450 nm, using a wavelength of 620 nm as a reference, and detected with a ThermoScientific Multiskan FC (Thermo Fisher Scientific). The graphs are representative results, and this experiment was independently repeated three times with similar results.

## 4. Conclusions

We developed a novel scaffold, diphenyl acrylonitrile, for promoting neurogenesis. Diphenyl acrylonitriles **29b**, **32a**, and **32b** showed significant neurogenesis-promoting activity. However, **29a**, the regioisomer of **32a**, did not show activity. The SARs studied indicated that the position of the cyano group attached to the olefin and the substituents installed onto the phenyl ring have significant impacts on the potency of promoting proliferation and differentiation. Excitingly, **32b** readily penetrated the BBB and exhibited better potency than **NSI-189** in elevating the neurogenesis level in rats’ hippocampi. The exposure of **32b** in brain tissue in terms of the AUC (i.e. the brain to plasma ratio, B/P) was 1.59 times higher than that in plasma (383 h*ng/mL in plasma and 608 h*ng/g in brain tissue). Additionally, the SARs also indicated that employing BrdU is considerably limited for predicting outcomes of neurogenesis, as many compounds that could promote proliferation in this work showed an inability to promote differentiation of newborn cells into mature neurons. 

Although the molecular target of **32b** is still unclear, this study showed which sections of the stilbene could tolerate structural modification, which is valuable for future designs for pulling down potential molecular targets.

## Data Availability

Data is contained within the article and Appendix A.

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
