# Peer review of "Discovery of Novel Diphenyl Acrylonitrile Derivatives That Promote Adult Rats’ Hippocampal Neurogenesis"

_ijms, 2024, doi:10.3390/ijms25021241_

Round 1
Reviewer 1 Report (Previous Reviewer 1)
Comments and Suggestions for Authors
All comments were adequetly addressed by the authors.
Reviewer 2 Report (Previous Reviewer 2)
Comments and Suggestions for Authors
Thank you for sending the revised manuscript and supplement. The authors made the corrections I asked for. The supplementary materials presents the spectra of the compounds described in this publication.
This manuscript is a resubmission of an earlier submission. The following is a list of the peer review reports and author responses from that submission.
Round 1
Reviewer 1 Report
Comments and Suggestions for Authors
The paper reports the synthesis of a series of compounds and the examination of their proliferative activity and further examination (of the most potent compounds) to ascertain their ability of promoting differentiation of newborn neural progenitor cells into mature neurons.
Although the investigation is limited to identifying a compound that could cross BBB and bring about the reported results, without investigating the transport route across BBB or any other molecular target to explain the obtained results, the propose of a chemical structure for further investigation is acknowledged. A few comments are the following:
Although there are several reports on the metabolic stability of nitriles, and the authors measured 32b in blood and brain, please clarify whether the drug that has been measured in the pharmacokinetic study is 32b (nitrile), or whether you (also) measured any other -possible- metabolite(s). The UPLC conditions (gradient; flow rate; wavelength; etc) used for these experiments are not reported. You can also attach the corresponding UPLC of 32b and those of the pharmacokinetic experiment, as a supplement, to provide the needed experimental results of this section of the paper.
The dose of 4.0 mg/kg that is used seems to be linked with the authors’ previous publication No 21. Please add this reference also in section 2.3., maybe here: “Previous in vitro or in vivo studies mainly focused on stimulating proliferation effect of the small molecules.” Regarding the dose - cytotoxicity evaluation, a dose dependent cytotoxicity set of experiments, at least for the most potent compound 32b, is of importance.
Also, evaluation of the cytotoxicity of the suggested compound using a model cell line for BBB (e.g. hCMEC/D3 cells) is considered essential for this kind of investigation.
Reviewer 2 Report
Comments and Suggestions for Authors
The publication "Discovery of Novel Stilbene Analogs that Promote Adult Rats' Hippocampal Neurogenesis" is interesting. It is concerned with the search for the new compound structures that promote adult rats' neurogenesis. The schemes of synthesis pathways of the final products are clearly presented. Unfortunately, the Authors did not present the spectra of these compounds.
1) In the line 100 there is an incorrect numbering of reactants, it should be written ... benzaldehyde (14) with phenyl ethanone (13), I suggest reversing the order of the compounds.
2) In the lines (535, 614) „under Argon protection” to under argon protection.
3) In the line 613 I think it should be written not "halo benzene", but better, for example: the derivative of the aromatic halogen.
4) In the line 99 there should be „aldol” condensation.
5) I think the group names for compound 8c should be N,N-dimethylaminocarbonyl and for compound 8d the cyclobutanocarboxy group (the line 151).
6) The spectra of the compounds are not attached. Please include spectra of the final products.
Reviewer 3 Report
Comments and Suggestions for Authors
The manuscript by Liu et al., "Discovery of novel stilbene analogs that promote adult rats’ hippocampal neurogenesis," presents an interesting set of results from the chemical synthesis of stilbene derivatives: 1,3-diphenyl 1-propene, 1,3-12 diphenyl 2-propene, 1,2-diphenyl acrylo-1-nitrile, 1,2-diphenyl acrylo-2-nitrile, 1,3-diphenyl trime-13 thylamine.
From a chemical synthesis viewpoint, it appears to be a well-accomplished, original work, having achieved an extensive collection of compounds that were tested in two experimental models: proliferation and differentiation of rat hippocampal cells. From the perspective of evaluating pharmacological activity, the work is limited in that only one dose of each drug under study was tested. However, as a first approach to a structure-activity relationship, it is a meritorious and interesting work.
Conceptually, the work is current and relevant. The search for drugs that can promote neurogenesis should be on the research agenda as it may have applicability in a variety of neurological diseases and in preventing neuronal loss with aging.
However, the manuscript in its current form has flaws that require additional work to be in a condition to be accepted for publication.
The title is not very informative. It tells the reader little about the main findings of the study. The introduction is entirely disjointed. The main ideas are there, but their sequencing is chaotic. It seems that there was a formatting change that swapped the order of the sentences.
The figures need significant improvements, particularly regarding the captions. The reader cannot understand what is shown, the grouping criteria of the compounds displayed in each figure (figures 2 - 5 and 7). Figures 6 and 8 need to be better explained. The captions should be more illuminating, and it should be explained to the reader how they the effects of the active and inactive drugs in the immunohistochemistry images can be seen.
On page 10, it is unclear why data on the partition coefficient of various compounds are placed in the section on pharmacokinetics when it is announced that the pharmacokinetic study will only be carried out with drug 32b.
The results presented in figures 2 to 5 refer to 8 slices from only two animals. How robust is an analysis of the effect in a group of only two animals?
The compound NSI-189 was used in the structure-activity assays as a control because it is known to be a neurogenesis stimulator (lines 239 - 240). As a positive control, shouldn't it be active? How do they explain that it is inactive both in proliferation (figure 7) and differentiation (figure 8)? Does the lack of response of a drug used as a positive control not call into question the entire methodology used?
In the description of the pharmacokinetic study, it is indicated that the animals were divided into eight groups. Why were the eight groups necessary? A more detailed explanation of the technical procedures is recommended.
Finally, the conclusions should be significantly improved. The conclusions can be inferred from the results, but the points selected by the authors as of most relevance from their study should be discussed in more detail.